# Diagnosis of brucellosis: Combining tests to improve performance

**Paul Loubet**[1☯], **Chloé Magnan**[2☯], **Florian Salipante**[3], **Théo Pastre**[2], **Anne Keriel**[4], **David O'Callaghan**[4], **Albert Sotto**[1], **Jean-Philippe Lavigne**[2]*

1 Virulence Bactérienne et Infections Chroniques, INSERM U1047, Univ Montpellier, Service des Maladies Infectieuses et Tropicales, Centre National de Reference des brucelloses, CHU Nîmes, Nîmes, France, 2 Virulence Bactérienne et Infections Chroniques, INSERM U1047, Univ Montpellier, Service de Microbiologie et Hygiène Hospitalière, Centre National de Reference des brucelloses, CHU Nîmes, Nîmes, France, 3 Department of Biostatistics, Epidemiology, Public Health and Innovation in Methodology (BESPIM), CHU Nîmes, Univ Montpellier, Nîmes, France, 4 Virulence Bactérienne et Infections Chroniques, INSERM U1047, Univ Montpellier, Centre National de Reference des brucelloses, Nîmes, France

☯ These authors contributed equally to this work.
* jean.philippe.lavigne@chu-nimes.fr

## Abstract

### Introduction

Brucellosis, a zoonotic infectious disease caused by bacteria of the genus *Brucella*, remains a significant global health concern in many parts of the world. Traditional diagnostic methods, including serological tests, suffer from limitations, including low sensibility and high false-positive rates, emphasizing the need for improved diagnostic strategies. In this study, we aimed to optimize diagnostic accuracy by reevaluating serological tests and exploring novel diagnostic algorithms.

### Methods

A retrospective observational study was conducted using sera collected between June 2012 and June 2023 at the French National Reference Center for *Brucella*. Various serological tests, including Rose Bengal plate test (RBT), standard agglutination test (SAT), Brucellacapt, and ELISA for IgM and IgG, were performed. Different diagnostic algorithms were evaluated, combining RBT with SAT, Brucellacapt, and ELISA to enhance the performance of diagnostic tests.

### Results

Among 3587 sera analyzed, 148 were confirmed cases of human brucellosis. Individual serological tests exhibited good sensitivity and specificity but lacked diagnostic accuracy. However, combining RBT with SAT or Brucellacapt significantly improved diagnostic performance, with reduced false positives. The most promising results were observed when an algorithm was built combining RBT, Brucellacapt, and ELISA for IgM and IgG (a score value of 0.5 with 90.5% for sensitivity, 99.7% for specificity, 92.4% for PPV, and 99.6% for NPV).

**Data Availability Statement:** The authors confirm that all data underlying the findings are fully available without restriction. All relevant data are

within the paper and its Supporting Information files.

**Funding:** The author(s) received no specific funding for this work.

**Competing interests:** The authors have declared that no competing interests exist.

## Conclusions

Serological tests remain crucial for brucellosis diagnosis, but their limitations necessitate innovative diagnostic approaches. Combining multiple serological tests in diagnostic algorithms shows promise in improving diagnostic accuracy. Efforts to refine diagnostic, strengthen surveillance, and raise awareness are essential for effective brucellosis control, particularly in resource-limited settings.

### Author summary

Brucellosis is a zoonotic infectious disease caused by bacteria of the genus *Brucella*. This disease remains a significant global health concern in many parts of the world. Traditional diagnostic methods include serological tests. However, these tests have major limitations due to their low sensibility and high false-positive rates. This poses a significant challenge in clinical practice, emphasizing the necessity for careful interpretation of test results and the development of more specific diagnostic tools. In this study, we assess the utility of traditional and easily accessible serological tests. Instead of using them individually, we decided to integrate the results of these multiple testing methods. We highlight the value of combining serological results into an algorithm to enhance the diagnosis of brucellosis and achieve greater overall accuracy in diagnosis. Efforts to streamline diagnostic, bolster surveillance, and increase awareness are essential for controlling brucellosis, particularly in resource-limited settings with a high disease burden.

## Introduction

Brucellosis, a zoonotic infectious disease caused by bacteria of the genus *Brucella*, remains a significant global health concern, particularly in regions where agriculture and animal husbandry are prevalent. Despite decades of research and control efforts, brucellosis continues to pose challenges due to its diverse clinical manifestations, high prevalence in certain populations (1.6–2.1 million new human cases annually), and potential for chronicity if left untreated [1–4].

*Brucella* is a facultative intracellular Gram-negative coccobacillus [5]. Among this genus, four species are responsible for human brucellosis: *B. melitensis* (sheep and goats), *B. suis* (domestic pigs), *B. abortus* (cattle), and *B. canis* (dogs) [1,4]. Humans can become infected by consuming contaminated food, mainly raw milk and dairy products, or by direct contact with infected animals or their bodily fluids, including through the aerosolization of contaminated samples in laboratories [4,6,7].

Brucellosis is a significant human health concern in many parts of the world, especially in Mediterranean countries, the Middle East, South and Central Asia, and Central and South America [6,8]. While some countries, such as France, are officially disease-free, cases still occur in individuals returning from endemic regions. The clinical presentation of brucellosis varies widely, ranging from acute febrile illness with nonspecific symptoms to chronic debilitating conditions affecting multiple organ systems [9]. This variability underscores the importance of accurate and timely diagnosis to facilitate appropriate treatment and prevent complications and relapse [10]. Definitive diagnosis of *Brucella* infection relies on bacterial

isolation, but culture sensitivity decreases as the disease progresses (i.e., during subacute and chronic phases) [8].

Traditional diagnostic methods for brucellosis include culture, serology, and molecular techniques [7,11]. The identification of human cases of brucellosis heavily depends on microbiological analysis, given the variability and lack of specificity of the associated symptoms of this disease [5]. While these approaches have been instrumental in identifying *Brucella* species and confirming infection, they often suffer from limitations such as low sensitivity, cross-reactivity with other pathogens, and the need for specialized laboratory facilities [7]. Due to their simplicity, cost-effectiveness, and high negative predictive value (NPV), serological tests remain the primary diagnostic tools in endemic areas with limited resources and in countries where the prevalence is low (and considered as a neglected disease) [11]. However, their results require interpretation that is often difficult and frequently inconclusive [12]. The 2006 WHO guidelines recommended the use of the Rose Bengal plate test (RBT) as a sensitive rapid screening test, with confirmation by other more specific tests such as standard agglutination test (SAT), ELISA, or microagglutination tests [13]. The RBT is a card agglutination test that detects both agglutinating and non-agglutinating antibodies, providing positive/negative results [11]. Studies have shown that using serum dilutions in positive qualitative RBT results can improve the test's specificity [14,15]. However, this test can yield false positive results, the presence of *Brucella* antibodies in individuals who do not have an active infection. Several factors contribute to false positives, including issues with hemolyzed serum, cross-reactivity with antibodies produced against other pathogens, prior exposure to *Brucella* or related organisms, and non specific binding of antibodies. In areas where brucellosis is rare, the positive predictive value (PPV) of serological tests may be compromised due to the low likelihood of true positives relative to false positives. This can lead to unnecessary treatment, increased healthcare costs, and patient anxiety. Furthermore, the sensitivity and specificity of serological tests may vary depending on factors such as the stage of infection, the immune status of the patient, and the strain of *Brucella* involved. False positives can also occur if tests are not performed or interpreted correctly, highlighting the importance of proper training and quality control measures in laboratory settings [11].

Strategies to reduce false positives include using dilution-based cut-offs for positivity, combining multiple tests to improve diagnostic accuracy, and incorporating clinical algorithms that integrate serological modalities. Moreover, commercialization of the serological kit for SAT, validated for human diagnostics, has recently been discontinued, making it necessary to identify new serological diagnostic solutions.

Ultimately, improved diagnostic tools and strategies are essential for effective disease management, surveillance, and control, thereby reducing the burden of brucellosis on human and animal populations worldwide.

In this study, our objectives were to (i) reevaluate the utility of RBT dilution for diagnosing of human brucellosis and (ii) assess the performance of diagnostic algorithms combining RBT and more specific tests.

## Methods

### Ethics statement

This study was conducted in accordance with the Declaration of Helsinki and had been reviewed and approved by the "Institutional Review Board" Ethics Committee of Nîmes University Hospital under the following number: 2024.03.05.

## Patients and human sera

This retrospective observational study was conducted in accordance with the Declaration of Helsinki and French law [16,17]. Sera sent to the French National Reference Center for *Brucella* (Department of Microbiology, Nîmes University Hospital, Nîmes, France) between June 2012 and June 2023 were included in the analyses. The local ethics committee (Institutional Review Board of Nîmes University Hospital; registration number n˚2024.03.05) approved their use in this research. Before inclusion, patients were informed of the study and their rights to oppose the use of their serology results. A verbal consent was obtained from the different patients.

Patients were classified based on the expertise of the National Reference Center: i) confirmed human brucellosis if they exhibited epidemiological and clinical findings and the positivity of blood/tissue culture or had negative cultures but serological tests positive (SAT titer ≥ 1/160; RBT positive; IgG/IgM >11 UI; Brucellacapt≥ 1/160); ii) negative in other situations (no epidemiological and clinical findings, no cultures and negative serological tests).

## Serological tests

All samples from a given patient were processed simultaneously and frozen at -20˚C.

For the titrated RBT, 30 μL of non-diluted serum was mixed with an equal volume of a suspension in an acidic medium (pH 3.65 ±0.05) and buffered commercial *Brucella abortus* antigen stained with Rose Bengal (BioMérieux, Marcy l'Etoile, France) [15]. The solution was previously equilibrated at room temperature and shaken to resuspend any bacterial sediment. The mixture was rocked at room temperature for 4 min according to the manufacturer's recommendations. Any visible agglutination and/or a typical rim were considered positive. Positive sera were titrated by serial twofold dilutions in physiological saline (1/2 to 1/256 dilutions), with 30 μL of dilutions being tested in the same manner as pure serum.

The Wright standard tube agglutination (STA) was performed using a commercial *B. abortus* antigen (issued from killed *B. abortus* 119–3 strain) and saline as the diluent (50 μL per well) in tube format (Brucella-WRIGHT, Bio-Rad, Marnes-la-Coquette, France) following the manufacturer's instructions. A large dilution panel (1/20 to 1/10240) of dilutions was tested to avoid negative results due to a prozone phenomenon. The SAT reactions were read after a 24-h incubation at 37˚C. The highest serum dilution showing >50% agglutination was considered the agglutinating titer [12]. A titer of 160 is currently considered the positivity threshold in non-endemic areas [11].

The Brucellacapt (Vircell, Granada, Spain) test employs an immunocapture agglutination technique to detect all classes of antibodies specific to *Brucella* species in serum samples. Brucellacapt tests were performed according to the manufacturer's instructions. Serum dilutions varying between 1/80 to 1/5120 were performed in strips of microtiter wells coated with anti-human immunoglobulins. Diluted patient serum and antigen were added to the wells, and the strips were incubated for 24-h at 37˚C. Positive reactions show agglutination over the bottom of the well, while negative reactions are indicated by a pellet at the center of the bottom of the well. A cut-off titer of >1/160 is currently used [18–20].

ELISAs for IgM and IgG antibodies were performed on each serum sample using a commercial kit (Vircell). The testing procedure followed the manufacturer's instructions. The test is performed in 96-well microtiter plates that are precoated with *Brucella* antigen. Titers over 11 U for IgG and IgM were considered positive.

## Statistical analyses

Patients with an "undetermined" diagnosis (< 2 positive serological tests and without any clear epidemiological or clinical data confirming brucellosis) and those with missing values for two

of the five tests performed were excluded from the analyses. Diagnostic performances of the different tests and predictive models under evaluation were assessed using sensitivity, specificity, NPV, and PPV. The Area under the ROC curve (AUC) was also reported.

All statistical analyses were conducted using R statistical software version 4.2.0. It should be noted that both quantitative and qualitative variables (the test results) are included in the model. RBT, SAT, and Brucellacapt values represent the dilution levels at which the results become positive. This information was considered a quantitative variable in the model after transforming negative results (samples that never became positive regardless of dilution) to a value of 2. Pure serum was considered as 1, a ½ dilution as 0.5, and so on. IgM and IgG ELISA tests were qualitative variables taking only positive/negative values. The best thresholds were selected to maximize the PPV while maintaining good sensitivity (close to 85% or greater).

Three algorithmic propositions were tried to enhance the diagnosis of brucellosis. The first proposition relies solely on the diluted RBT to improve specificity. The second proposition combines the diluted RBT and SAT or Brucellacapt results. A logistic regression model was employed to integrate these variables and enhance diagnostic performance. The third proposition incorporates four serological tests (RBT, Brucellacapt and IgM and IgG ELISA), also utilizing logistic regression. The formulas for the different models are provided in the article; the indicator function, denoted as $1_{\{x = a\}}$, yields a value of 1 if the condition within the braces is true (x = a here) and 0 otherwise.

For each score, a ROC curve was generated, and optimal thresholds were selected based on optimized sensitivity and specificity. Diagnostic performances, including AUC, sensitivity, specificity, PPV, and NPV, were presented along with their confidence intervals (95%CI). Given that predictive models were constructed, the data were partitioned into a training set (2/3) and a test set (1/3) while maintaining a balance in disease status. Performances were computed in the test set, but final models (leading to the provided formulas) were derived using the entire dataset. Performance metrics for the entire dataset were also provided and demonstrated similarity to those of the test set.

## Results

### Sera analyzed

Among the 3674 sera received at the French National Reference Center for brucellosis during the study period, 13 were excluded due to missing at least two of the test results (due to insufficient serum quantity), and 74 were excluded due to an undetermined status of the serology. Among the remaining 3587 sera definitively included, 148 were considered as confirmed human brucellosis cases, and 3439 were negative (**Fig 1**).

### Performance of serological tests

The performance of the five serological tests used in this study is presented in **Table 1**. Overall, these tests exhibited excellent specificities (94.4% to 99.3%) and good sensitivities (72.4% to 97.3%). However, the poor PPVs observed (42.6% to 84%) indicated that the different serological tests needed to be more suitable for drawing clear conclusions on the diagnosis.

Among the serological tests, the SAT is considered as the first screening test. However, as we have seen, it is no longer commercially available in Europe. Interestingly, in countries with low brucellosis prevalence, it is recommended to use a threshold of 1/160 to define test positivity. However, in our series conducted on 2105 sera (due to the discontinuation of SAT commercialization), the test performed better at a titer of 1/80 (sensitivity 95.1%; specificity 90.9%) compared to a titer of 1/160 (sensitivity 86.0%; specificity 94.9%).

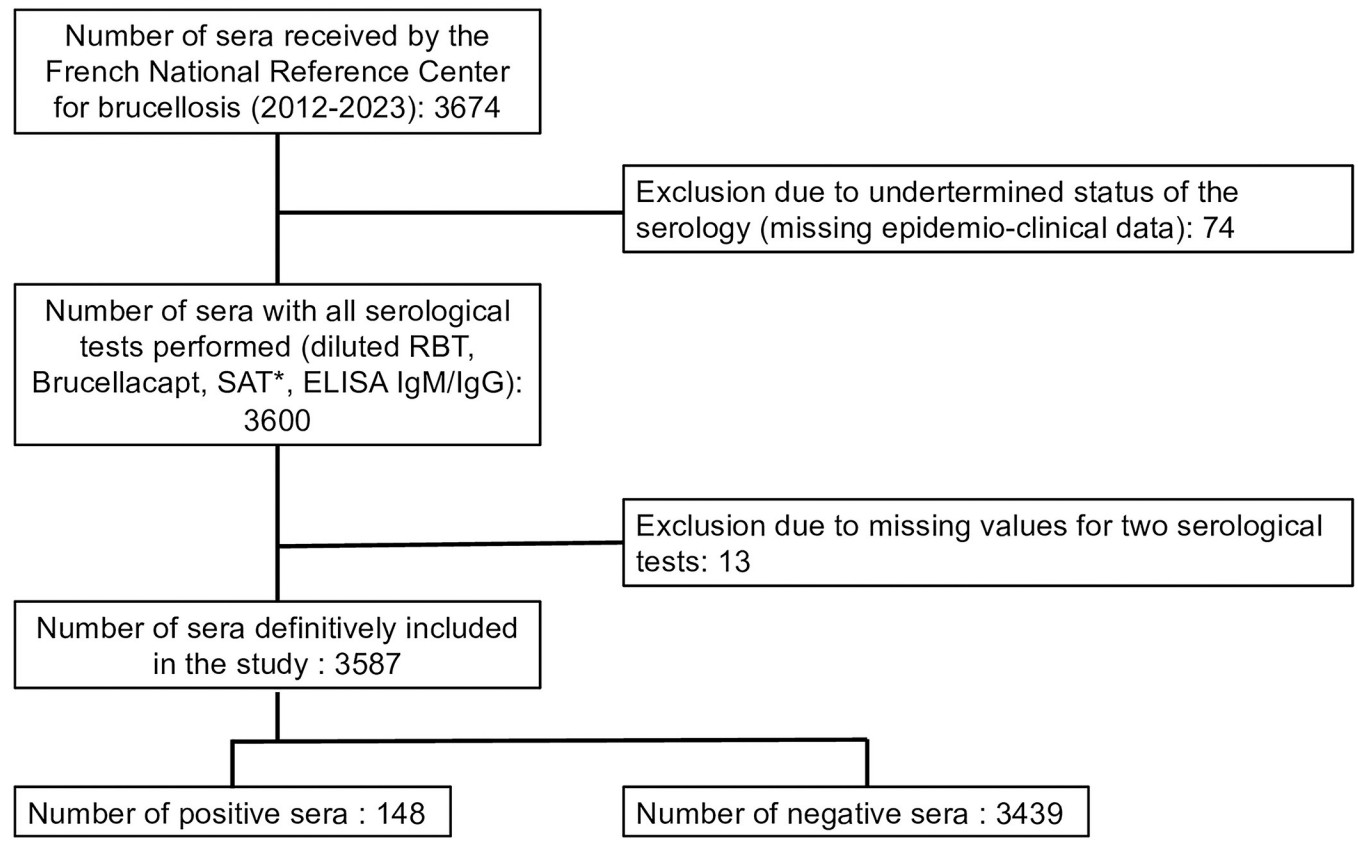

**Fig 1. Flowchart of the study.** *Evaluation was exclusively performed on 2105 sera due to the discontinuation of the kit's commercialization.

## Performance of diluted RBT

When performed on pure serum, the sensitivity of RBT was 97.3% [93.2%, 99.3%], whereas the PPV was only 42.6% [37.3%, 48.1%]. When we defined a limit titer > ¼, we observed better performance with 85.1% (95%CI [78.4%, 90.4%]) for sensitivity, 99.3% [99.0%, 99.6%] for specificity, 84.0% [77.1%, 89.5%] for PPV, and 99.4% [99.0%, 99.6%] for NPV (**Table 2**). However, 22 positives were not detected using this limit, but the number of false positives was reduced from 194 to 24.

**Table 1. Performance of serological tests used to diagnose brucellosis in an area of low prevalence.**

|  | Sensitivity % (95%CI) | Specificity % (95%CI) | VPP % (95%CI) | NPP % (95%CI) |
|---|---|---|---|---|
| RBT (threshold: pur) | 97.3 (93.2–00.3) | 94.4 (93.5–95.1) | 42.6 (37.3–48.1) | 99.9 (99.7–100) |
| SAT (threshold 1/160)* | 86.0 (79.2–91.2) | 94.9 (93.8–95.8) | 55.2 (48.4–61.8) | 98.9 (98.4–99.3) |
| Brucellacapt (threshold 1/160) | 82.4 (75.3–88.2) | 99.3 (99.0–99.6) | 84.1 (77.2–89.7) | 99.2 (98.9–99.5) |
| IgM ELISA (threshold > 11U) | 73.8 (65.8–80.7) | 99.2 (98.9–99.5) | 80.5 (72.7–86.8) | 98.9 (98.5–99.2) |
| IgG ELISA (threshold > 11U) | 72.4 (64.4–79.5) | 99.2 (98.8–99.4) | 78.4 (70.4–85.0) | 98.8 (98.4–99.2) |

*Evaluation was exclusively performed on 2105 sera due to the discontinuation of the kit's commercialization.

**Table 2. The performance of the diluted Rose Bengal test (RBT) was used to diagnose brucellosis.**

| Dilution threshold (Positive if) | Sensitivity (%) | Specificity (%) | PPV (%) | NPV (%) |
|---|---|---|---|---|
| Pure serum | 97.3% | 94.4% | 42.6% | 99.9% |
| >1/2 | 94.6% | 97.2% | 59.6% | 99.8% |
| >1/4 | 85.1% | 99.3% | 84.0% | 99.4% |
| >1/8 | 67.6% | 99.8% | 94.3% | 98.6% |
| >1/16 | 43.9% | 99.9% | 97.0% | 97.6% |

*PPV, positive predictive value; NPV, negative predictive value

## Performance of diagnostic algorithm combining RBT and SAT

Firstly, we combined the two main serological tests, RBT and SAT results, to determine the best diagnostic algorithm for diagnosing brucellosis. We established the following formula where RBT and SAT corresponded to the positive dilution titer for a given patient:

$$Score_{RBT+SAT} = \frac{\exp(1.936 - 4.614 \times \textbf{RBT} - 1.451 \times \textbf{SAT})}{(1 + \exp(1.936 - 4.614 \times \textbf{RBT} - 1.451 \times \textbf{SAT}))} *$$

\* RBT and SAT are the dilutions from which the test becomes positive. A negative result takes the value 2.

Using a threshold of 0.5, the average performances in 10-fold cross-validation were 84% (Standard Deviation (SD):12%) for sensitivity, 99% (SD: 1%) for specificity, and an AUC = 0.99 (SD: 0.02).

On the whole data set, the performances are 85.3% [78.4%, 90.7%] for sensitivity, 99.1% [98.6%, 99.5%] for specificity, 87.1% [80.4%, 92.2%] for PPV, and 98.9% [98.4%, 99.3%] for NPV. The AUC of the association of these two tests was 0.989 [0.98,0.997] (**Table 3**). In this model, 16 sera were false positive (on the 1962 negative sera). Due to the cessation of SAT commercialization, this score was established only on 2105 sera.

On the test set, the performances are 87.2% [74.3%, 95.2%] for sensitivity, 98.8% [97.7%, 99.5%] for specificity, 83.7% [70.3%, 92.7%] for PPV, and 98.8% [97.7%, 99.5%] for NPV. The AUC of the association of these two tests was 0.991 [0.984, 0.998].

## Performance of diagnostic algorithm combining RBT and Brucellacapt

We then evaluated the association between diluted RBT and Brucellacapt, establishing the following formula:

$$Score = \frac{\exp(0.1615 - 3.8422 \times \textbf{RBT} + 2.6457 \times 1(\textbf{Brucellacapt} = \textbf{positive}))}{(1 + \exp(0.1615 - 3.8422 \times \textbf{RBT} + 2.6457 \times 1(\textbf{Brucellacapt} = \textbf{positive})))}$$

\* RBT and Brucellacapt are the dilutions from which the test becomes positive. A negative result takes the value 2.

Using a threshold of 0.5, the average performances in 10-fold cross-validation were 83% (SD:13%) for sensitivity, 100% [SD: 1%] for specificity, and an AUC = 0.98 (SD: 0.03).

On the whole data set, the performances are 83.1% [76.1%, 88.8%] for sensitivity, 99.5% [99.2%, 99.7%] for specificity, 88.5% [82%, 93.3%] for PPV, and 99.3% [98.9%, 99.5%] for NPV. The AUC of the association of these two tests was 0.986 [0.974,0.998]. In this model, 43 were false positives (out of 3439 negative sera).

**Table 3. Comparison of the performance of different diagnostic algorithms using various combinations of serological test results to diagnose brucellosis.**

|  | Threshold (positive if) | Sensitivity % | Specificity % | PPV % | NPV % | No. of false positives | No. of positives diagnosed/No. total of positives |
|---|---|---|---|---|---|---|---|
| Diluted RBT | 0.25 | 85.1 | 99.3 | 84.0 | 99.4 | 24 | 126/148 |
| RBT + SAT* | 0.5 | 85.3 | 99.1 | 87.1 | 98.9 | 18 | 122/143 |
| RBT + Brucellacapt | 0.5 | 83.1 | 99.5 | 88.5 | 99.3 | 16 | 123/148 |
| RBT + Brucellacapt + ELISA IgM and IgG | 0.5 | 90.5 | 99.7 | 92.4 | 99.6 | 11 | 134/148 |

*Evaluation was exclusively performed on 2105 sera due to the discontinuation of the kit's commercialization

On the test set, the performances are 83.7% [70.3%, 92.7%] for sensitivity, 99.2% [98.5%, 99.6%] for specificity, 82% [68.6%, 91.4%] for PPV, and 99.3% [98.6%, 99.7%] for NPV. The AUC of the association of these two tests was 0.995 [0.991, 0.999].

### Performance of diagnostic algorithm combining all serological tests

Finally, we combined all serological results to evaluate a diagnostic algorithm using the following formula:

$$y = 0.8754 - 2.9266 \times \textbf{RBT} - 1.7714 \times 1_{\{IgG=negative\}} + 1.0885 \times 1_{\{IgG=positive\}}$$
$$- 0.5704 \times 1_{\{IgM=negative\}} + 0.5149 \times 1_{\{IgM=positive\}} + 2.2159 \times 1_{\{Brucellacapt=Positive\}}*$$

$$\textbf{Score} = \frac{\exp(y)}{(1 + \exp(y))}$$

* In this formula, the symbol "1" followed by brackets on index referred to the indicator function. The indicator function takes the value 1 if the condition specified into braces is fulfilled and 0 otherwise. RBT corresponds to the dilution from which the test becomes positive. A negative result takes the value 2.

The average performances in 10-fold cross-validation were 89% (SD: 9%) for sensitivity, 100% (SD: 1%) for specificity, and an AUC = 0.99 (SD: 0.03).

On the whole data set, the performances are 90.5% [84.6%, 94.7%] for sensitivity, 99.7% [99.4%, 99.8%] for specificity, 92.4% [86.8%, 96.2%] for PPV, and 99.6% [99.3%, 99.8%] for NPV. The AUC of the association of these two tests was 0.997 [0.995, 0.999]. In this model, 11 were false positives (out of 3439 negative sera).

On the test set, the performances are 95.9% [86%, 99.5%] for sensitivity, 99.2% [98.5%, 99.6%] for specificity, 99.2% [98.5%, 99.6%] for PPV, and 99.8% [99.4%, 100%] for NPV. The AUC of the association of these two tests was 0.997 [0.994, 0.999].

### Discussion

Brucellosis poses significant diagnostic challenges, complicating public health efforts due to the broad extent of individuals' contact with infected animals or their products. The nonspecific symptoms associated with brucellosis, such as fever and joint pain, often lead to misdiagnosis or underdiagnosis [11]. Healthcare providers may overlook or misinterpret the disease, delaying appropriate treatment. Moreover, diagnostic inaccuracies are prevalent in regions with limited access to advanced diagnostic tools and healthcare infrastructure, resulting in underestimated disease burden and inadequate public health responses. Paradoxically, diagnostic difficulties can lead to the overestimation of brucellosis cases in regions reliant on less

specific diagnostic methods [7,21]. Therefore, there is an urgent need to improve diagnostic methods.

Serological tests, although lacking sensibility and sometimes yielding challenging results in individuals repeatedly exposed to *Brucella* organisms, remain a cornerstone in diagnosis, particularly in resource-limited countries [11,22]. They are also important tools in countries where brucellosis has been eradicated and the disease has been 'forgotten'. Evaluating the serodiagnosis tests for brucellosis is complex due to the absence of a single gold standard test against which all other laboratory assays can be measured. Combining existing and cost-effective tests presents an interesting early diagnosis and treatment solution. In this study, we examined the diagnostic potential of five tests for detecting exposure to *Brucella*: the RBT, SAT, Brucellacapt, and ELISA tests for IgM and IgG. RBT, known for its wide-spread use as a screening test, often yields a weak PPV due to the often low prevalence of brucellosis and its high sensitivity, particularly as serum dilutions are not usually tested. To address this, our study aimed to optimize test performance while maintaining sensitivity and reducing false positives by establishing dilution-based cut-offs for positivity. We explored various scenarios, from individual test considerations to combinations aimed at enhancing performance.

Utilizing dilutions in RBT gave a significant improvement in diagnostic accuracy, highlighting the importance of this approach in refining testing protocols as previously reported [14,15]. By diluting serum samples, nonspecific reactions can be reduced, enhancing test specificity, leading to more reliable results, and ultimately improving diagnostic accuracy. Furthermore, our investigation into combining tests underscores the potential for enhancing diagnostic capabilities in the detection of anti-*Brucella* antibodies. Integrating multiple testing methods allows us to use the strengths of each, potentially achieving greater overall accuracy in diagnosis. This integrated approach represents a promising avenue in clinical management, demonstrating the potential benefits of a comprehensive testing strategy. The effectiveness of combining RBT with SAT has already been established, particularly where these tests are the primary diagnostic options available [11]. This underscores the importance of maximizing diagnostic capabilities with the resources at hand. However, due to the discontinuation of SAT commercialization in Europe, it was essential to establish a new strategy to enhance diagnostic accuracy and improve patient care. Finally, combining RBT and Brucellacapt with ELISA IgM and IgG shows a very promising performance, drastically limiting false positives and suggesting the potential for enhanced diagnostic accuracy with a multi-test approach. To our knowledge, this approach is original and has never been reported. However, further investigation is warranted to determine the reliability of this combined algorithm. Specifically, evaluating the algorithm's performance beyond just the AUC is crucial, as other metrics such as sensitivity, specificity, and PPV provide a more comprehensive assessment of diagnostic accuracy.

Despite their numerous drawbacks [23] such as low sensibility and the difficulty in interpreting results [12,21]), serological tests remain an indispensable diagnostic tool for human brucellosis. Traditionally, the principle was to use a screening test to evoke the diagnosis, followed by a confirmatory test [24]. Here we show that combining tests can improve performance and represents a promising avenue for enhancing diagnostic capabilities in brucellosis detection. By integrating multiple testing methods and combining the strengths of each, such as the sensitivity of one test and the specificity of another, we can achieve greater overall accuracy in diagnosis. This approach demonstrates the potential benefits of a comprehensive and integrated testing approach in managing brucellosis. Research efforts focused on improving the performance of serological tests and developing novel diagnostic techniques are also underway.

## Acknowledgments

The authors thank the Nîmes University Hospital for its structural, human, and financial support through the award obtained by our team during the internal call for tenders 'Thématiques phares'. We also thank Sarah Kabani for her editing assistance.

## Author Contributions

**Conceptualization:** Jean-Philippe Lavigne.

**Data curation:** Paul Loubet, Chloé Magnan, Théo Pastre, Anne Keriel.

**Formal analysis:** Paul Loubet, Chloé Magnan, Florian Salipante, Albert Sotto, Jean-Philippe Lavigne.

**Funding acquisition:** David O'Callaghan, Jean-Philippe Lavigne.

**Investigation:** Paul Loubet, Chloé Magnan, Théo Pastre, Anne Keriel, David O'Callaghan, Jean-Philippe Lavigne.

**Methodology:** Florian Salipante, Jean-Philippe Lavigne.

**Project administration:** Albert Sotto, Jean-Philippe Lavigne.

**Software:** Florian Salipante, Théo Pastre.

**Supervision:** David O'Callaghan, Albert Sotto.

**Validation:** Paul Loubet, Chloé Magnan, Jean-Philippe Lavigne.

**Visualization:** David O'Callaghan, Albert Sotto, Jean-Philippe Lavigne.

**Writing – original draft:** Paul Loubet, Chloé Magnan, Jean-Philippe Lavigne.

**Writing – review & editing:** Florian Salipante, Théo Pastre, Anne Keriel, David O'Callaghan, Albert Sotto.

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
