## [Decision Letter · Decision Letter 0]

18 May 2024

Dear Dr. Lavigne,

Thank you very much for submitting your manuscript "Diagnosis of brucellosis: combining tests to improve performance" for consideration at PLOS Neglected Tropical Diseases. As with all papers reviewed by the journal, your manuscript was reviewed by members of the editorial board and by several independent reviewers. The reviewers appreciated the attention to an important topic. Based on the reviews, we are likely to accept this manuscript for publication, providing that you modify the manuscript according to the review recommendations. 

Sincerely,

Joseph M. Vinetz

Section Editor

Joseph Vinetz

Section Editor

Reviewer's Responses to Questions

**Key Review Criteria Required for Acceptance?**

**Methods**

-Are the objectives of the study clearly articulated with a clear testable hypothesis stated?

-Is the study design appropriate to address the stated objectives?

-Is the population clearly described and appropriate for the hypothesis being tested?

-Is the sample size sufficient to ensure adequate power to address the hypothesis being tested?

-Were correct statistical analysis used to support conclusions?

-Are there concerns about ethical or regulatory requirements being met?

Reviewer #1: The authors have applied Rose Bengal plate test (RBT), Wright standard tube agglutination (STA), Brucellacapt® (Vircell, Granada, Spain) test, and/or ELISA for IgM and IgG antibodies to determine whether the combination of these tests will improve the clinical diagnosis of brucellosis. They have included 3,587 serum samples in the test. The authors have performed RBT, RBT+SAT, RBT+Brucellacapt®, or RBT + Brucellacapt® + ELISA IgM and IgG tests with statistical analysis. There are no concerns about ethical or regulatory requirements.

**Results**

-Does the analysis presented match the analysis plan?

-Are the results clearly and completely presented?

-Are the figures (Tables, Images) of sufficient quality for clarity?

Reviewer #1: The authors have applied all these techniques and found that 148 serum samples are positive, white 3,439 serum samples are negative. The results, including the figures and Tables are well presented.

**Conclusions**

-Are the conclusions supported by the data presented?

-Are the limitations of analysis clearly described?

-Do the authors discuss how these data can be helpful to advance our understanding of the topic under study?

-Is public health relevance addressed?

Reviewer #1: This is a straightforward seroloigcal study. The authors have concluded that the combination of multiple serological tests of brucellosis may improve diagnostic accuracy, which may help effective brucellosis control, particularly in resource-limited settings.

**Editorial and Data Presentation Modifications?**

Reviewer #1: A minor modification is needed. Please see the enclosed revised manuscript.

**Summary and General Comments**

Reviewer #1: The manuscript didn't show any novel test. However, the proposed test may help to improve the serologic test to improve the diagnosis of brucellosis.

PLOS authors have the option to publish the peer review history of their article (what does this mean?). If published, this will include your full peer review and any attached files.

Reviewer #1: No

Figure Files:

Data Requirements:

Reproducibility:

References

---

## [Editor Report · Decision Letter 1]

3 Jun 2024

Dear Dr. Lavigne:

Thank you very much for submitting your manuscript "Diagnosis of brucellosis: combining tests to improve performance" (PNTD-D-24-00551R1) for review by PLOS Neglected Tropical Diseases. 

As with all papers reviewed by the journal, your manuscript was reviewed by members of the editorial board and by several independent reviewers. Based on the reviews, we regret that we will not be pursuing this manuscript for publication at PLOS Neglected Tropical Diseases.

This is primarily a review and this topic has been very much addressed already. This ms does not advance the field.

The reviews are attached below this email, and we hope you will find them helpful if you decide to revise the manuscript for submission elsewhere.

While we cannot consider your manuscript further for publication in PLOS Neglected Tropical Diseases, we would like to offer you the option to transfer your submission, with reviews, to PLOS Global Public Health https://www.editorialmanager.com/PGPH/

If you DO wish to transfer your submission, please click this link:

<DeepLinkData><DeepLinkTypeID>27</DeepLinkTypeID><peopleID>149726</peopleID><userSecurityID>ed463a46-7cfc-4daf-8502-24ce222a9620</userSecurityID><documentID>33474</documentID><revision>1</revision><manuscriptNumber>PNTD-D-24-00551</manuscriptNumber><docSecurityID>af32d043-7d85-4c7f-85a8-0ed555ed5407</docSecurityID></DeepLinkData>

If you do NOT wish to transfer your submission, please click this link to decline:

<DeepLinkData><DeepLinkTypeID>28</DeepLinkTypeID><peopleID>149726</peopleID><userSecurityID>ed463a46-7cfc-4daf-8502-24ce222a9620</userSecurityID><documentID>33474</documentID><revision>1</revision><manuscriptNumber>PNTD-D-24-00551</manuscriptNumber><docSecurityID>af32d043-7d85-4c7f-85a8-0ed555ed5407</docSecurityID></DeepLinkData>

Please note, all PLOS journals are editorially independent and vary in submission requirements.

Should you choose to transfer, your manuscript files, along with the reviewers' comments and their identities will be transferred automatically, and you will receive a confirmation email within 24 hours. Once transferred, your submission will be returned to you so you can check over your record before completing the submission. You may be asked to provide additional information, such as a response to the reviewers' comments. If you have any questions, please contact the editorial office of PLOS Global Public Health https://www.editorialmanager.com/PGPH/

We are sorry that the news is not more positive on this occasion, and we hope you will consider PLOS Neglected Tropical Diseases for future submissions. Thank you for your support of PLOS and of open-access publishing.

Sincerely,

Joseph M. Vinetz

Section Editor

Joseph Vinetz

Section Editor

Shaden Kamhawi

co-Editor-in-Chief

orcid.org/0000-0003-4304-636X

Paul Brindley

co-Editor-in-Chief

---

## [Editor Report · Decision Letter 2]

7 Aug 2024

Dear Dr. Lavigne,

We are pleased to inform you that your manuscript 'Diagnosis of brucellosis: combining tests to improve performance' has been provisionally accepted for publication in PLOS Neglected Tropical Diseases.

Best regards,

Joseph M. Vinetz

Section Editor

Joseph Vinetz

Section Editor

---

## [Editor Report · Acceptance letter]

30 Aug 2024

Dear Dr. Lavigne,

We are delighted to inform you that your manuscript, "Diagnosis of brucellosis: combining tests to improve performance," has been formally accepted for publication in PLOS Neglected Tropical Diseases.

Best regards,

Shaden Kamhawi

co-Editor-in-Chief

Paul Brindley

co-Editor-in-Chief
